# Investigation on Preparation and Performance of High Ga CIGS Absorbers and Their Solar Cells

**DOI:** 10.3390/ma16072806

**Published:** 2023-03-31

**Authors:** Xiaoyu Lv, Zilong Zheng, Ming Zhao, Hanpeng Wang, Daming Zhuang

**Affiliations:** 1Faculty of Materials and Manufacturing, Beijing University of Technology, Beijing 100124, China; lxy4518@emails.bjut.edu.cn (X.L.); zilong.zheng@bjut.edu.cn (Z.Z.); 2School of Materials Science and Engineering, Tsinghua University, Beijing 100084, China; zhaoming2013@tsinghua.edu.cn (M.Z.); wanghp20@mails.tsinghua.edu.cn (H.W.)

**Keywords:** CIGS, Ga content, CuIn/CuGa precursors, tandem solar cell, band gap energy

## Abstract

Tandem solar cells usually use a wide band gap absorber for top cell. The band gap of CuIn_(1−x)_Ga_x_Se_2_ can be changed from 1.04 eV to 1.68 eV with the ratio of Ga/(In+Ga) from 0 to 1. When the ratio of Ga/(In+Ga) is over 0.7, the band gap of CIGS absorber is over 1.48 eV. CIGS absorber with a high Ga content is a possible candidate one for the top cell. In this work, CuInGa precursors were prepared by magnetron sputtering with CuIn and CuGa targets, and CIGS absorbers were prepared by selenization annealing. The Ga/(In+Ga) is changed by changing the thickness of CuIn and CuGa layers. Additionally, CIGS solar cells were prepared using CdS buffer layer. The effects of Ga content on CIGS thin film and CIGS solar cell were studied. The band gap was measured by PL and EQE. The results show that using structure of CuIn/CuGa precursors can make the band gap of CIGS present a gradient band gap, which can obtain a high open circuit voltage and high short circuit current of the device. With the decrease in Ga content, the efficiency of the solar cell increases gradually. Additionally, the highest efficiency of the CIGS solar cells is 11.58% when the ratio of Ga/(In+Ga) is 0.72. The value of Voc is 702 mV. CIGS with high Ga content shows a great potential for the top cell of the tandem solar cell.

## 1. Introduction

As environmental issues become increasingly serious, people focus on the decomposition of pollutants [1,2,3] and the use of clean energy [4,5,6]. Photovoltaic materials can convert solar energy into electric energy, which can make full use of solar energy. Chalcopyrite phase CuIn_(1−x)_Ga_x_Se_2_ (CIGS) based solar cell has been the focus of research due to its high absorption coefficient and adjustable band gap and high potential efficiency. With the increase in Ga content, the band gap of CIGS varies from 1.04 eV to 1.67 eV [7]. The highest efficiency of CIGS solar cells obtained with the ratio of Ga/(In+Ga) (GGI) of 0.30 is 23.35% [8]. Thus, great efforts have been made for high efficiency as a single p-n junction CIGS solar cell almost with GGI of 0.2 to 0.3. There are few studies on CIGS with high GGI. Nowadays, tandem solar cells have attracted considerable attention [9,10,11]. Thus, CIGS with high Ga content is considered to be a suitable material for top cells of CIGS based tandem solar cells due to its high band gap [12,13,14,15]. M. Schmid et al. prepared simply stacked CuGaSe_2_/Cu(In,Ga)Se_2_ tandem cell and achieved a efficiency of 8.5% [16]. The efficiency of CGS top cell is only 4.3%. They thought that the top cell performs an important role in the tandem solar cell. Hedayati M et al. obtained 32.3% efficiency cell by using CGS/CIGS tandem solar cells through simulation, and the top cell used is CGS solar cell with 16% efficiency [17]. Thus, a top cell with high quality is necessary for tandem solar cells. Additionally, CIGS with high GGI is a promising candidate for the top cells in tandem systems.

In the multi-compounds system, the system of the similar bonding and varying stoichiometry can be assessed by systematic atomistic modeling, which can explain the variation of their specific properties [18]. In the CuInGaSe_2_ system, the band gap varies with the different ratio of Ga/(In+Ga). Through the hybrid density functional theory study, the conduction band of CIGS consists by In, Ga, and Se s orbitals. With the increasing of GGI, the density of the states from In s orbitals becomes lower, thus, the band gap of CIGS increases [19]. With the increase in Ga content, the band gap of CIGS solar cells can be calculated by [20]
E_g_(eV) = 1.04 + 0.63x − 0.21x(1 − x)(1)
where x is the Ga/(In+Ga) ratio. Samira Khelif et al. calculated that the optimal band gap range of the top cell of the top cell is 1.5–1.7 eV [21]. For CIGS with Ga content from 0.7 to 1.0, the band gap can be changed between 1.45 eV to 1.67 eV. With the increase in the band gap, a higher open circuit voltage (Voc) can be obtained. However, the short circuit current (Jsc) would decrease because of the lower absorption limit. Therefore, the double-graded band gap of CIGS absorbers can enhance their photovoltaic performance [22,23]. A high band gap at the surface results in a high Voc. Meanwhile, a lower band gap minimum can improve the Jsc of the absorber.

CIGS solar cells with high Ga content are mainly prepared by evaporation [24,25], electrodeposition [26] and molecular beam epitaxy [27], and magnetron sputtering and selenide post-treatment [28,29]. Among these preparations, magnetron sputtering and selenide post-treatment processes are considered to have the advantage of large area uniformity [30]. However, CIGS films fabricated by sputtering CIGS quaternary have small grains, which may limit the efficiency of the solar cells [31,32]. Therefore, using CuGa and CuIn target sputtering, and selenide post-treatment is used to prepare CIGS solar cells. The precursor prepared with CuGa and CuIn alloy targets were annealed in a selenium-containing atmosphere, which can make the grains grow fully [33]. Additionally, the ratio of GGI can be adjusted easily.

In this work, CuInGa precursors were prepared by magnetron sputtering with CuIn and CuGa targets, and CIGS absorbers were prepared by selenization annealing, and the physical properties of CIGS films were investigated. At the same time, CIGS solar cells were prepared, and the effects of different Ga content on the performance of solar cells were investigated as well.

## 2. Materials and Methods

The structure of CIGS solar cell fabricated in this work was SLG/Mo/CIGS/CdS/i-ZnO/AZO. CuIn and CuGa precursors were deposited on Mo-coated soda-lime glass (SLG) substrates by middle-frequency magnetron sputtering from Cu_0.8_In and Cu_0.8_Ga alloy targets. First, 800 nm Mo was prepared on SLG by magnetron sputtering. In order to make the precursor uniform, we used a target with a larger size of 360 × 80 × 6 mm, the sputtering background pressure was 2 × 10^−3^ Pa, the sputtering gas was Ar, and the pressure was 0.7 Pa. Using CuIn/CuGa as the precursor, the CuIn layer was below the CuGa layer. The content of Ga in the precursor was changed by changing the sputtering thickness of CuIn and CuGa precursors. The designed values of GGI were 0.7, 0.8, 0.9, and 1.0, respectively. The total thickness of the precursor was designed as 500 nm. The prepared precursor subjected to selenization annealing in a vacuum annealing furnace. In order to stabilize and fully conduct the reaction, two steps of temperature rise were used for annealing. The base pressure of the annealing furnace was 2 × 10^−3^ Pa. The temperature of the first step is 450 °C, last for 30 min; The temperature of the second stage is 540 °C, last for 40 min. The mixture of hydrogen selenide and argon was used for selenization, and the concentration of hydrogen selenide was 3%. The thickness of the CIGS films was about 1 μm. The CdS buffer layer was prepared by the CBD method. The i-ZnO and AZO window layers were prepared by magnetron sputtering, with thickness of 80 nm and 600 nm, respectively.

The inductively coupled plasma optical emission spectrometer (ICP-OES, IRIS Intrepid II, Thermo Fisher Scientific, Waltham, MA, USA) was used to measure Ga concentration in CIGS films. The morphologies of CIGS thin films were observed by SEM (Merlin VP Compact, Zeiss, Jena, Germany). The phase structures were characterized by X-ray diffraction (XRD, D/max-2550, Rigaku, Tokyo, Japan). The surface band gap was measured by PL, the 532 nm laser line was used for photoluminescence (PL) excitation. The device performance parameters of CIGS cells were characterized by a Volt-Ampere characteristic tester (2400, Keithley, Cleveland, OH, USA) and a solar simulator (91,192–1000, Newport, Oxfordshire, UK) under AM 1.5G illumination at 25 °C. The external quantum efficiency spectra of CIGS preparation solar cells were measured by a QETest Station 2000ADI system (Crowntech, Hong Kong).

## 3. Results

### 3.1. Results and Discussion

#### 3.1.1. Cu(In,Ga)Se_2_ Growth and Characterization

Table 1 shows the specific components of CIGS absorption layer with different Ga contents. The contents of Cu, In, Ga, and Se were obtained by ICP, Ga/(In+Ga) are 1.00, 0.90, 0.81, and 0.72, respectively. Figure 1 shows SEM images of the surface and cross-sectional morphologies of CIGS films with Ga/(In+Ga) ratio of 1.00, 0.90, 0.81, and 0.72, respectively, on Mo-coated SLG substrate.

According to the surface images, all samples show good crystallinity and grains are closely arranged. With the increase in indium content, holes begin to appear on the surface of the absorption layer and, at the same time, the planeness of the surface decreases. This is because of the structure of the precursor. CuIn layer was deposited at the bottom half and CuGa layer was deposited at the upper half. In the H_2_Se atmosphere, indium showed more effective than gallium, resulting in the decrease in the flatness [34,35]. The blue line is for the distribution of selenium along the depth of the CIGS film. In the samples with GGI of 1.00 and 0.90, the content of Se near Mo layer decreases significantly, forming a layer with high Ga content. This is because the reaction rate of Cu-Se combination is higher than that of Ga-Se, which makes Ga aggregate near Mo layer, resulting in a layer with high Ga content that is difficult to selenide at Mo/CIGS surface. The red line is for the distribution of indium along the depth of the CIGS film. With the increase in indium content, the combination reaction between Cu-In-Ga-Se is more likely to occur, reducing the tendency of enrichment of Ga toward Mo. When Ga content is 0.81, Se can be evenly distributed in the CIGS film. For Sample B, Sample C, and Sample D, the distribution of indium is increased first, at the depth of about 300 nm, indium content reaches the maximum value. It means that at the depth of 300 nm, the CIGS film has a minimum band gap. As the depth increases, the indium content decreases.

The PL method was used to obtain the band gap of the surface of different CIGS films. Figure 2 shows the photoluminescence spectrum measured by PL. When GGI is 1.00, the film is CuGaSe_2_, and the band gap is 1.66 eV, which is consistent with the reported value [29]. With the increase in indium content, the surface band gap gradually decreases. The values of the band gaps are 1.59 eV, 1.55 eV, and 1.48 eV, when GGI are 0.90, 0.81, and 0.72, respectively. Table 2 shows the surface GGI measured by ICP-OES and the band gaps calculated by Equation (1). The band gaps of the surface are slightly higher than that calculated from ICP. In the preparation of CIGS by selenization after sputtering, Ga is easy to enrich on the back surface, result in a lower GGI at the surface, leading to a decrease in the open circuit voltage [36]. For the CuIn/CuGa precursor in this work, after annealing the band gap of the surface is higher than that of the average, which is helpful for the improvement of the open circuit voltage.

The XRD pattern and lattice constants of a and c are shown in Figure 3. All samples show good crystallinities, and the grains mainly grow along (1 1 2) direction. The diffraction peaks at 2θ of near 27.77°, 45.74°, 46.23°, 54.33°, and 55.15° correspond to the (1 1 2), (2 2 0), (2 0 4), (3 1 2), and (1 1 6) planes of CGS phase (# 75-0104), respectively. The (2 2 0) and (2 0 4) reflections are split at the room temperature, which is due to the tetragonal distortion in CIGS [37]. With the increase in Ga content, the peak position gradually shifts to the right, and the strength of the high angle (2 0 4) peak gradually increases. When Ga completely replaces In, the peak strength of the (2 0 4) peak increases significantly. This is because the atomic radius of Ga is smaller than In. When Ga^3+^ replaces In^3+^, the lattice size decreases [24]. The estimated lattice constants a and c in a unit of CIGS crystal were increased from 5.603 Å and 10.974 Å to 5.655 Å and 11.109 Å, respectively, with a decreasing GGI ratio from 1 to 0.72. This is because with the decrease in GGI, more In^3+^ with a larger ionic radius of 0.8 Å replaced Ga^3+^ with that of 0.62 Å. The variation of a and c for CIGS with GGI from 0.7 to 1.0 can be expressed as the following equations:(2)ax=−0.188x+5.787
(3)cx=−0.506x+11.478

The peaks of (1 1 2) planes of Sample A to Sample D are 27.77°, 27.65°, 27.51°, and 27.46°, respectively. Combining with the (1 1 2) position of CuInSe2(26.67°, #80-0535), the GGI in the CIGS film can be calculated. For Sample B and Sample C, the ratios of Ga/(In+Ga) calculated by XRD are 0.88 and 0.80, respectively, which are smaller than the results detected by ICP. It is might due to a part of Ga atoms do not enter the crystal phase [33]. For Sample D, the ratio of Ga/(In+Ga) calculated by XRD is 0.72, which is equal to the result detected by ICP, it means that almost all of Ga atoms enter the crystal phase.

#### 3.1.2. Characterization of CIGS Solar Cells

The solar cell J-V curve of CIGS cells using CdS as buffer layer is shown in Figure 4a. With the increase in indium content, solar cell efficiency gradually increases. Table 3 lists the performance parameters of different samples. When GGI is 0.72, the efficiency of the solar cell achieves the highest value of 11.58%. With the increase in GGI, the band gap of the absorption layer increases, which decreases the spectral response range, and the short circuit current of the cell decreases significantly. When GGI is 0.72, the highest value of Jsc is 25.2 mA/cm^2^. When GGI is 0.81, the open circuit voltage reaches the highest value. This is because when GGI is 1.00 and 0.90, a layer of high Ga content and poor Se content is formed at the Mo/CIGS surface. At the same time, a thin fault appears between this layer and CIGS layer, which intensifies the carrier recombination at the fault and affects the open circuit voltage, resulting in a high deficit of Voc. With GGI continuing to decrease, the layer at Mo/CIGS surface is replaced by CIGS grains and the fault disappears. Therefore, the open circuit voltage returns to the normal level, and decreases with the decrease in the surface band gap. The open circuit voltage reaches the highest value of 757 mV when GGI is 0.81. With the increase in GGI, the band gap of CIGS increases, which is mainly reflected in the increase in conduction band minimum (CBM) [19]. When GGI = 1.00, a “cliff like” is formed between the absorption layer CGS and the buffer layer CdS, which intensifies the carrier recombination at the CGS/CdS surface and reduces Voc and FF [38,39]. When GGI increases, the conduction band offset (CBO) between CIGS and CdS increases, so the filling factor decreases with the increase in GGI. Larsson F et al. used ZTO as the buffer layer and gained a CGS solar cell with high Voc of 1017 mV, which was much higher than CdS buffer (768 mV). Additionally, FF increased from 58.6% to 68.0% [24]. Therefore, changing a suitable buff layer should be considered in future work.

Figure 4 shows (b) the EQE curve of solar cells with different GGI, (c) the minimum band gap obtained from the EQE curve. Table 4 shows the comparison between surface band gap and minimum band gap in the CIGS absorbers. Among all absorbers with different GGI, the minimum band gap in the absorbers is smaller than the surface band gap.

According to the results of EDS, PL, and EQE, the Eg depth profile of CIGS absorbers can be obtained. The sketch of band gap of the CIGS absorber is shown in Figure 5. At the surface, the band gaps were measured by PL, the band gaps of Sample B, Sample C, and Sample D are 1.59 eV, 1.55 eV, and 1.48 eV, respectively, the indium content increases with depth. At the depth of about 300 nm, indium content reaches the maximum value. Meanwhile, the CIGS film has a minimum band gap which was measured by EQE. The values are 1.49 eV, 1.37 eV, 1.34 eV, respectively. As shown in Figure 1, with the depth continuing to increase, the indium content begins to decrease, causing the increase in GGI, the band gap increases. The band gap of CIGS film presents like a V-shaped. A wider band gap at the surface makes the solar cell obtain a high open-circuit voltage; additionally, a lower band gap minimum makes the solar cell have a high short-circuit current [40].

## 4. Conclusions

The properties of CuInGaSe_2_ thin films and solar cells with different Ga contents were investigated. The results showed that using the structure of CuIn/CuGa as a precursor, a double-graded band gap profile can be obtained after selenidation annealing, which can make the solar cell have high open circuit voltage and high short circuit current. With the decrease in GGI, the solar cell efficiency increases gradually. The highest efficiency obtained in this work is 11.58% at GGI of 0.72. When GGI is 0.81, the maximum open circuit voltage of 757 mV obtained. CIGS solar cells with wide band gap show the potential to be suitable materials for top cells of CIGS tandem solar cells.

## Figures and Tables

**Figure 1 materials-16-02806-f001:**
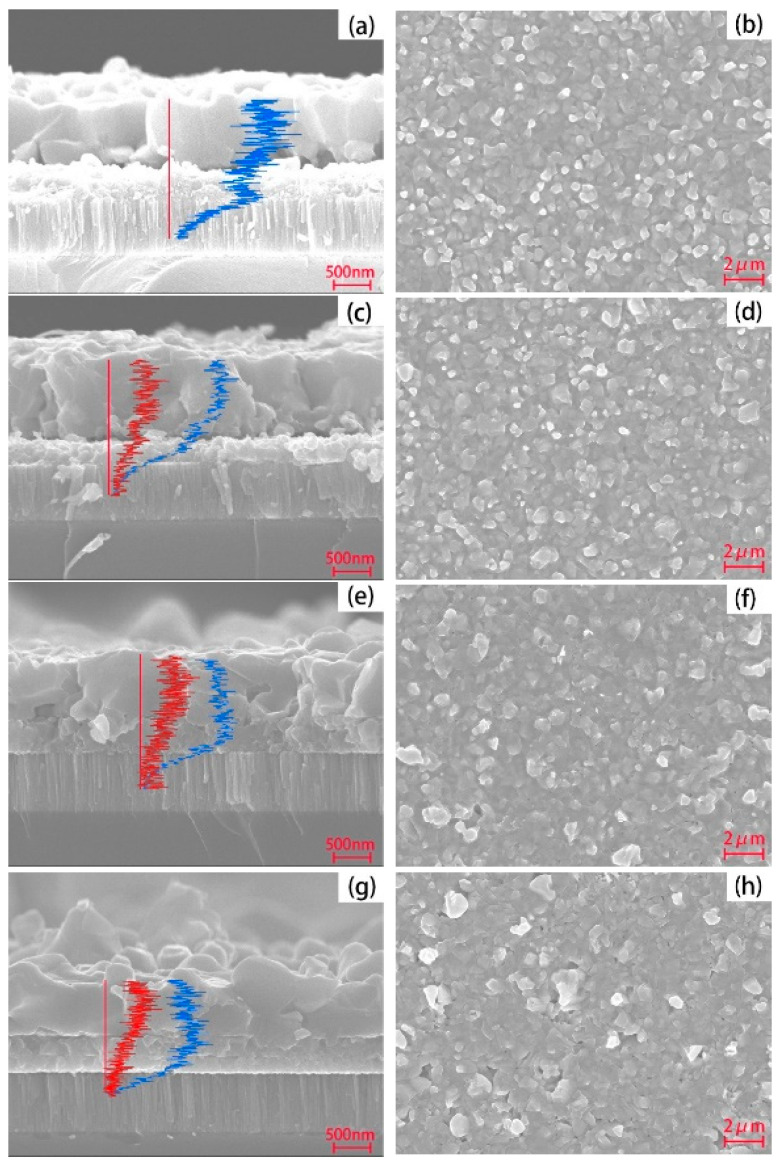
Cross-sectional and surface SEM images: (**a**,**b**) for GGI = 1.00; (**c**,**d**) for GGI = 0.90; (**e**,**f**) for GGI = 0.81; (**g**,**h**) for GGI = 0.72. The red line is for the distribution of indium along the depth of the CIGS film, the blue line is for the distribution of selenium along the depth of the CIGS film.

**Figure 2 materials-16-02806-f002:**
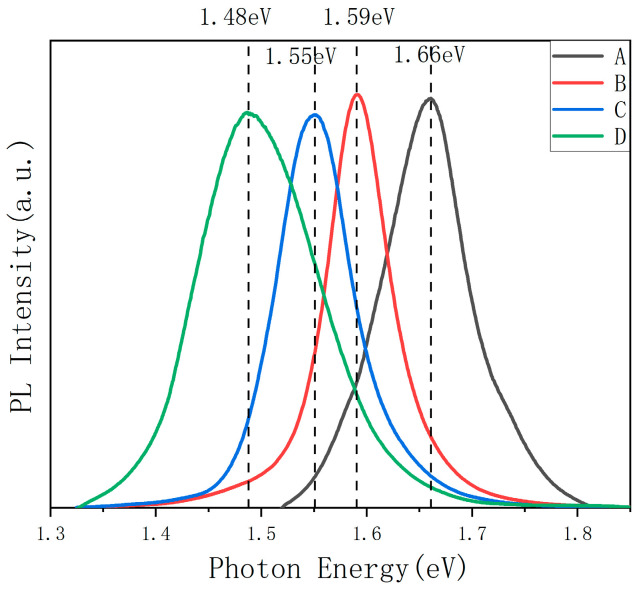
The PL spectra of the CIGS films with different gallium content.

**Figure 3 materials-16-02806-f003:**
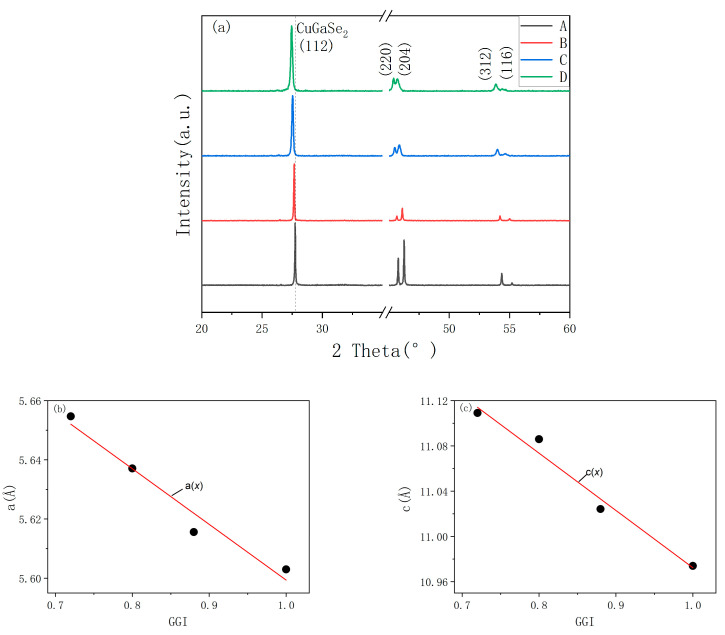
(**a**) The XRD spectra of the CIGS films with different gallium content, (**b**,**c**) lattice constants of a and c calculated from (**a**).

**Figure 4 materials-16-02806-f004:**
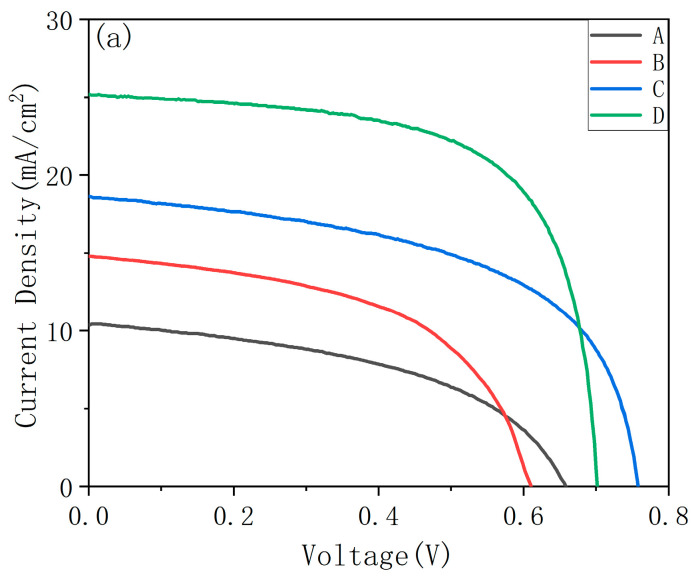
(**a**) The J-V curve of CIGS cells using CdS as buffer layer, (**b**) their corresponding EQE spectra, and (**c**) the minimum band gap obtained from the EQE curve.

**Figure 5 materials-16-02806-f005:**
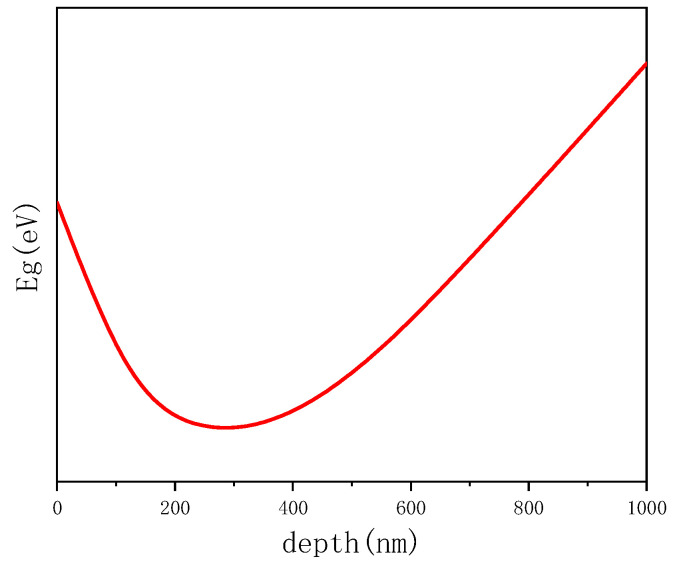
The sketch of Eg depth profile of CIGS absorber.

**Table 1 materials-16-02806-t001:** Elemental compositions of the CIGS films.

Sample	Cu (at%)	In (at%)	Ga (at%)	Se (at%)	Ga/(In+Ga)
A	22.8	0	25.1	52.1	1.00
B	22.6	2.4	22.7	52.3	0.90
C	22.6	4.8	20.5	52.1	0.81
D	22.4	6.9	18.2	52.5	0.72

**Table 2 materials-16-02806-t002:** The Ga/(In+Ga) measured by ICP and the band gap measured by PL.

	Sample A	Sample B	Sample C	Sample D
Ga/(In+Ga) measured by ICP	1.00	0.90	0.81	0.72
Calculated E_g_(eV)	1.66	1.59	1.52	1.45
E_g_ measured by PL (eV)	1.66	1.59	1.55	1.48

**Table 3 materials-16-02806-t003:** Performance parameters of CIGS solar cells.

Sample	Ga/(In+Ga)	Efficiency (%)	Voc (mV)	Jsc (mA/cm^2^)	FF (%)	Voc,def (mV)
A	1.00	3.26	659	10.5	46.86	1001
B	0.90	4.76	610	14.8	52.70	980
C	0.81	7.79	758	18.6	55.20	792
D	0.72	11.58	702	25.2	65.55	778

**Table 4 materials-16-02806-t004:** The band gap of CIGS measured by EQE and PL.

Sample	Eg_min_ (eV)	Eg by PL (eV)
A	1.66	1.66
B	1.49	1.59
C	1.37	1.55
D	1.34	1.48

## Data Availability

Any further details relevant to this study may be obtained from the authors upon a reasonable request.

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
