# Peer review of "Investigation on Preparation and Performance of High Ga CIGS Absorbers and Their Solar Cells"

_materials, 2023, doi:10.3390/ma16072806_

Round 1
Reviewer 1 Report
Please refer attachment.

Reviewer 2 Report
This manuscript reports on the influence exercised by the Ga content on the performance of wide band gap CuInGaSe2 solar cells. In this work as per state-of-the-art in this particular task, CuIn and CuGa precursors are prepared by magnetron sputtering, while CuIn(1-x)GaxSe2 (CIGS) absorption layers are realized by selenide annealing. Tailoring the structure and properties by modification of the Ga content in CIGS thin film and CIGS becomes clearly an attractive approach for increased efficiency of the corresponding solar cell, and more so in the present case of synthesis method. The authors convincingly make the case for achieving high efficiency of CIGS with high Ga content top cell of the tandem solar cell-
The presentation of results is clear and appealing, an easy and consistent read and thus can be very helpful for the interested community to make choices and benchmarking when future attempts on improving the CIGS system for solar cells are made. Useful and employable future developments and solutions in relation to this approach can be extracted from the present results, while they are well transmitted by the short discussion in the manuscript. Growth and characterization (SEM, ICP, PL) are well described and highly adequate to the present task.
All in all, this work certainly represents a valuable contribution with possible wider impact in the field.
The authors chose an adequate structure of the manuscript. Also, they provided concise and nicely illustrated figures and their corresponding analysis.
The present manuscript is a significant contribution, this work once published would be instructive and suggestive in terms of further studies and with good chances be cited.
There are some relatively minor issues with this already excellent manuscript that will need to be addressed before the manuscript becoming suitable for publication, i.e., it can be considered for publication after a minor revision:
1: Title: It would be better if the word “influence” is used in singular, not in plural.
2: Authors should mention specific characterization techniques in the abstract. It would advertise their work much better.
3: The issue of the thermal stability (and the possible influence of varying Ga content) of the CIGS system is not discussed. Is there any relevant quantitative data related to this issue?
4: In the introduction, the authors miss that In and Ga containing systems of similar bonding and varying stoichiometry (e.g., but not only Ga) can be assessed by systematic atomistic modeling which can theoretically guide their synthesis and/or explain variation of their specific properties (including stability, band gap etc.) [Dalton Transactions 44 (2015) 3356-3366], an aspect that should be acknowledged also in this manuscript.
5: Spell-check and stylistic revision of the paper are necessary. Some long sentences, as well as misspellings, etc., are noticeable throughout the text.
Round 2
Reviewer 3 Report
Please, accept this article in present form.
Author Response
Thanks again to you for your suggestions to further improve this manuscript.